

# TP53 mutation and MET amplification in circulating tumor DNA analysis predict disease progression in patients with advanced gastric cancer

Jia Li[1,2,*], Zhaoyan Li[3,*], Yajie Ding[1], Yan Xu[1], Xiaohong Zhu[1], Nida Cao[1], Chen Huang[4], Mengmeng Qin[1], Feng Liu[1] and Aiguang Zhao[1]

[1] Department of Oncology, Longhua Hospital, Shanghai University of Traditional Chinese Medicine (TCM), Shanghai, China

[2] Department of Integrated Chinese and Western Medicine, Affiliated Cancer Hospital of Zhengzhou University and Henan Cancer Hospital, Zhengzhou, Henan, China

[3] Department of Oncology, Yueyang Hospital of Integrated Traditional Chinese and Western Medicine, Shanghai University of Traditional Chinese Medicine (TCM), Shanghai, China

[4] Department of General Surgery, Renji Hospital, Shanghai Jiao Tong University School of Medicine, Shanghai, China

[*] These authors contributed equally to this work.

Corresponding author
Aiguang Zhao, aiguangzhao@qq.com

## ABSTRACT

**Background**. Gastric cancer (GC) is a heterogeneous disease that encompasses various molecular subtypes. The molecular mutation characteristics of circulating tumor DNA (ctDNA) in advanced gastric cancer (AGC), especially the clinical utility of TP53 mutation and MET amplification in ctDNA need to be further explored.

**Objectives**. The aim of this study was mainly to assess the clinical utility of TP53 mutation and MET amplification in ctDNA as biomarkers for monitoring disease progression of AGC.

**Patients and Methods**. We used multigene NGS-panel technology to study the characteristics of ctDNA gene mutations and screen the key mutant genes in AGC patients. The Kaplan-Meier method was used to calculate the survival probability and log-rank test was used to compare the survival curves of TP53 mutation and MET amplification in ctDNA of AGC patients. The survival time was set from the blood test time to the follow-up time to observe the relationship between the monitoring index and tumor prognosis.

**Results**. We performed mutation detection on ctDNA in 23 patients with AGC and identified the top 20 mutant genes. The five most frequently mutated genes were TP53 (55%), EGFR (20%), ERBB2 (20%), MET (15%) and APC (10%). TP53 was the most common mutated gene (55%) and MET had a higher frequency of mutations (15%) in our study. Kaplan-Meier analysis showed that patients with TP53 mutant in ctDNA had shorter overall survival (OS) than these with TP53 wild ($P < 0.001$). The Allele frequency (AF) of TP53 mutations in patient number 1 was higher in the second time (0.94%) than in the first time (0.36%); the AF of TP53 mutations in patient number 16 was from scratch (0~0.26%). In addition, the AF of TP53 mutations in patients who survive was relatively low ($P = 0.047$). Simultaneously, Kaplan-Meier analysis showed that patients with MET amplification also had shorter OS than these with MET without amplification ($P < 0.001$).
**Conclusion**. TP53 and MET are the two common frequently mutant genes in ctDNA of AGC patients.TP53 mutation and MET amplification in ctDNA could predict disease progression of AGC patients.

# INTRODUCTION

Gastric cancer is one of the most common malignancies in the world. There are one million new cases of gastric cancer and 783,000 deaths each year in the world (*Bray et al., 2018*). There are about 679,000 new cases of gastric cancer and the number of deaths is about 498,000 each year in China (*Chen et al., 2016*). Despite advances in surgery-based comprehensive treatment, more than 30% patients with gastric cancer are diagnosed at an advanced stage and about 50% of patients after surgery will relapse and metastasize (*Peng et al., 2020*; *Zhao et al., 2019*; *Zong et al., 2016*). Therefore, over 80% of gastric cancer patients are in the advanced stage with poor clinical outcome in china. Due to the fact that gastric cancer is a heterogeneous disease that encompasses various molecular subtypes, personalized-medicine approaches that commonly rely on a single tumor-biopsy sample to portray tumor mutational landscapes cannot achieve satisfactory treatment effect (*Hu et al., 2012*; *Wong et al., 2014*; *Gerlinger et al., 2012*). It is urgently needed to circumvent the issues of intratumoral heterogeneity and the limitations and risks related to the biopsy procedure.

Non-invasive circulating tumor DNA (ctDNA) sequencing, a method of liquid biopsy, provides a potential tool for real-time monitoring of tumor disease progression (*Wang et al., 2019*). The ctDNA includes small fragments of double-stranded DNA (160–180 base pairs) and carries tumor related genetic alterations released by apoptotic and necrotic tumor cells (*Diaz & Bardelli, 2014*). In addition, ctDNA levels are also related to tumor burden (*Hamakawa et al., 2015*). Many studies have described the use of ctDNA in advanced tumor, including advanced gastric cancer (AGC) patients (*Marzese, Hirose & Hoon, 2013*; *Creemers et al., 2017*; *Frankell & Smyth, 2019*). *Hamakawa et al. (2015)* reported that ctDNA may serve as a useful biomarker to monitor gastric cancer progression and residual disease. *Wu et al. (2020)* also found that ctDNA could be used as an effective tool to monitor the efficacy of chemotherapy and predict prognosis in advanced gastric cancer. *Yang et al. (2020)* demonstrated that ctDNA could detect molecular residual disease and monitor for disease recurrence in definitively-treated locoregional gastric cancer. With the development of multigene NGS-panel technology, targeted capture sequencing can cost-effectively detect gene regions to provide cancer-associated mutations in ctDNA, thereby enhancing the clinical implementation of mutation analysis (*Meyerson, Gabriel & Getz, 2010*; *Metzker, 2010*). Recent studies using NGS have shown that a wide range of potential cancer-driven

genes have caused mutations in gastric cancer (*Cai et al., 2019*; *Chen et al., 2015*; *Xu et al., 2014*) .

TP53 is the most common mutated gene in gastric cancer, accounting for about 50% (*Bellini et al., 2012*). TP53 has an important role in regulating cell proliferation processes and maintaining genomic integrity and stability (*Ozaki & Nakagawara, 2011*; *Lane & Levine, 2010*). MET is a proto-oncogene located on chromosome 7q21-31 and encodes a hepatocyte growth factor (HGF) receptor and acts as a carcinogenic driver in tumors, resulting in extensive downstream effects on tumor growth, invasiveness, angiogenesis, epithelial-mesenchymal transition and metastasis (*An et al., 2014*; *Gherardi et al., 2012*; *Kim et al., 2018*).The relationship between TP53 and MET gene abnormalities (such as mutations and amplifications) and clinical pathology in gastric cancer tissues has been studied (*Baniak et al., 2016*; *Fenoglio-Preiser et al., 2003*; *Peng et al., 2014*). In addition, studies have reported that characteristics of genetic mutations in ctDNA and tumor tissue are inconsistent (*Creemers et al., 2017*; *Frankell & Smyth, 2019*; *Maron et al., 2019*). Therefore, the molecular mutation characteristics of ctDNA in AGC and the clinical utility of TP53 mutation and MET amplification in ctDNA need to be further explored.

In this study, we used multigene NGS-panel technology to study the characteristics of ctDNA gene mutations in AGC patients. Then, we analyzed the clinical utility of TP53 mutation and MET amplification in ctDNA of AGC patients.

## MATERIAL AND METHODS

### Patients and samples

The 23 patients included in this study were all patients with stage IV advanced gastric cancer. Patients were all drawn from Longhua Hospital, Shanghai University of Traditional Chinese Medicine. This study was approved by the ethics committee of shanghai university of traditional chinese medicine (2017LCSY021) and was consistent with the Helsinki Declaration. All patients received written informed consent for this study. The peripheral blood plasma samples of 23 patients were collected, of which 5 patients collected two blood samples. Therefore, a total of 28 blood samples were used to detect somatic mutations. Clinical characteristics of 23 AGC patients was also collected (Supplemental Material 1). The somatic mutations, somatic copy number alteration and clinical data of 224 AGC tissue samples were downloaded from the TCGA portal (https://tcga-data.nci.nih.gov/tcga/).

### DNA extraction and target capture sequencing

According to the manufacturer's instructions, plasma DNA were extracted from blood samples by QIAamp DNA Blood Mini Kit (Qiagen, Hilden, Germany). Later, the concentration of plasma DNA was measured by Qubit Fluorometer and Qubit dsDNA HS Analysis Kit (Invitrogen, Carlsbad, CA, USA). Then, the fragment status was evaluated by Agilent 2200 TapeStation system and the DNA Integrity Number (DIN) was generated by ScreenTape assay (Agilent Technologies, Santa Clara, CA, USA). Finally, DNA was sheared prior to using an ultrasonoscope with a peak of 250 bp, followed by end repair.

Indexed Illumina NGS libraries were prepared from circulating DNA libraries were prepared using the KAPA Library Preparation Kit (Kapa Biosystems, Wilmington,

MA, USA). The panel for detection of ctDNA in peripheral blood plasma samples from patients, were designed to cover 471 hot-spot mutations of 197 genes (including 17 NCCN guidelines-recommended tumor-associated genes and 180 common tumor burden-associated genes).

Next, Target enrichment was performed using the customized SeqCap EZ library (Roche NimbleGen, Madison, Wis.) according to the manufacturer's protocol. DNA sequencing was performed with $2 \times 150$ bp paired-end reads on the HiSeq 3000 sequencing system (Illumina, Inc). The genes of panel are listed in Supplemental Material 2.

## Identification of somatic mutation in ctDNA

Terminal adaptor sequences and low-quality reads were removed from the raw data. The clean reads were aligned to the human genome build GRCh37 using BWA software version 0.7.12-r1039. Somatic single nucleotide variants (SNVs) and small insertions and deletions (Indels) were generated using MuTect version 1.1.4 and GATK version 3.4-46-gbc02625, respectively. CONTRA v2.0.8 was used to detect copy number variants (CNVs). The candidate variants were all manually verified in the Integrative Genomics Viewer (IGV). Candidate somatic mutations were SNVs and Indels where the variant allele fraction (VAF) was ≥2% and there were ≥5 high-quality reads (Phred score ≥30, mapping quality ≥13, and without paired-end reads bias) containing the target base. Non-synonymous mutations annotated by ANNOVAR were used in clonal structure reconstruction.

## Research process analysis

To screen for common mutations in patients with advanced gastric cancer, we first performed NGS sequencing analysis on plasma samples from 23 patients with advanced gastric cancer and obtained gene sequencing data. Then, we compared the obtained sequencing data with the genetic sequencing data of patients with advanced gastric cancer in TCGA. Finally, we analyzed the TP53 gene mutation and MET gene amplification, and obtained the significance of TP53 and MET in the gene mutation data of advanced gastric cancer. The flow chart of this study was shown in Fig. 1.

## Statistical analysis

The mutation landscape across a cohort, including SNVs, InDels and mutational burden, were created by Genomic Visualizations in R (GenVisR). Kaplan–Meier survival plots were generated for TP53 mutation and MET amplification at baseline using log-rank tests. All statistical analyses were performed with SPSS (v.22.0, IL, Chicago, USA) or GraphPad Prism (v. 6.0, GraphPad Software, Chicago, USA) software. All statistical tests were bilateral and $P < 0.05$ was considered statistically significant.

## Result

## Characteristics of patients and sequencing data

All 23 AGC patients included in this study, and their clinical characteristics were presented in Table 1. Fifteen of them were older than 60 years and 20 were male. All the pathological types were adenocarcinoma, with 7 cases of liver metastasis and 4 cases of peritoneal metastasis. According to lauren classification (*Tang et al., 2020*) , 5 cases were classified as diffuse-type and 18 other types. Of these patients, 12 had undergone radical surgery, 4

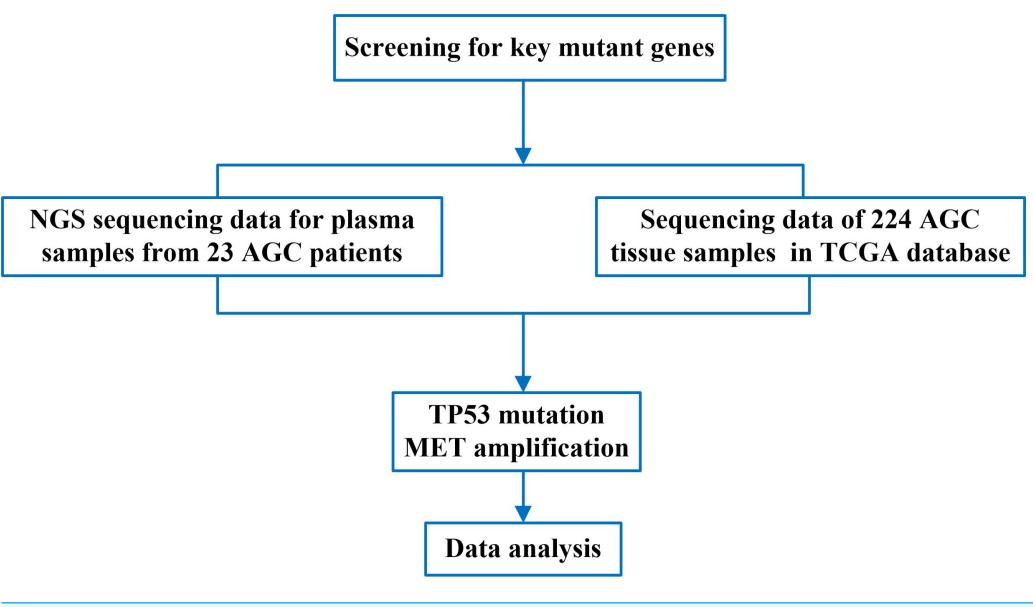

**Figure 1** The flow chart of this study.

had palliative resection, and 7 had no surgery. The cohort included 11 patients receiving chemotherapy and 12 patients receiving supportive care only. When performing gene sequencing, 20 patients had tumor burden. NGS sequencing results of 23 AGC patients were included in Supplemental Material 3.

## Landscape of somatic mutations

We performed mutations of ctDNA in 23 plasma samples from AGC patients and identified the top 20 mutant genes (Fig. 2A). The five most frequently mutated genes were TP53 (55%), EGFR (20%), ERBB2 (20%), MET (15%) and APC (10%). Figure 2B shows the top 20 mutant genes of tumor tissue from 224 AGC patients in the TCGA database. TP53 was the most common mutated gene in TCGA (44%) and our study (55%). MET amplification had a higher frequency of mutations (15%) in this study, while the frequency of mutations was very low in the TCGA database.

## Relationship between TP53 mutation and prognosis of patients with advanced gastric cancer

In the TCGA database, based on the survival analysis of the TP53 status (TP53 mutant and wild type), the overall survival (OS) clinically was not statistically significant. As shown in Figs. 3A and 3B, Kaplan–Meier survival analysis according to the TCGA database showed that TP53 status in early stage gastric cancer (stage I, II) and advanced gastric cancer (stage III, IV) was not associated with OS ($P = 0.57$ and $P = 0.14$).

In the analysis of ctDNA for our 23 plasma samples, the prognostic value of TP53 mutations was demonstrated. Kaplan–Meier analysis showed that patients with TP53 mutant had shorter survival time than these with TP53 wild (Fig. 3C, $P < 0.001$). Since the period of follow-up was set from the blood test time to the end of follow-up (2018.9.30),

**Table 1  The characteristics of 23 AGC patients detected by ctDNA.**

| Variables | Number (%) |
|---|---|
| **Age** | |
| <60 | 8 (34.8) |
| ≥60 | 15 (65.2) |
| **Sex** | |
| Female | 3 (13.0) |
| Male | 20 (87.0) |
| **Lauren classification** | |
| Diffuse | 5 (21.7) |
| Others | 18 (78.3) |
| **Liver metastasis** | |
| Yes | 7 (30.4) |
| No | 16 (69.6) |
| **Peritoneal metastasis** | |
| Yes | 4 (17.4) |
| No | 19 (82.6) |
| **Surgical approach** | |
| Radical surgery | 12 (52.2) |
| Palliative resection | 4 (17.4) |
| Others | 7 (30.4) |
| **Chemotherapy** | |
| Yes | 11 (47.8) |
| No | 12 (52.2) |
| **Tumor burden** | |
| Yes | 20 (87.0) |
| No | 3 (13.0) |

its significance indicated the relationship between the monitoring index (TP53) and the tumor prognosis.

To illustrate the role of TP53 mutation in predicting disease progression, we observed 5 AGC patients with rapid disease progression, and performed ctDNA sequencing on their blood samples again. The time of blood sample collection was the time of imaging examination or clinically judged as disease progression. TP53 mutations were detected in only 3 patients, namely patients 1, 16, and 17. No TP53 mutation was detected in the other two patients, patients 9 and 11. Then, we compared the results of the second ctDNA sequencing of patients 1, 16, and 17 with the first sequencing results, and mainly analyzed the changes in the Allele Frequency (AF) of TP53. We found that the AF of TP53 mutations in patient 1 was higher in the second time (0.94%) than in the first time (0.36%); the AF of TP53 mutations in patient 16 was from scratch (0∼0.26%). Moreover, the AF of TP53 in patient 17 ranged from 0 to 0.17%, which is similar to that of patient 16 (Fig. 3D). It indicated that the AF of TP53 mutations was positively correlated with tumor progression. Although TP53 mutations had also occurred in patients who survive, it was clearly found that AF was relatively low (Fig. 3E, $P = 0.047$).

Peer J

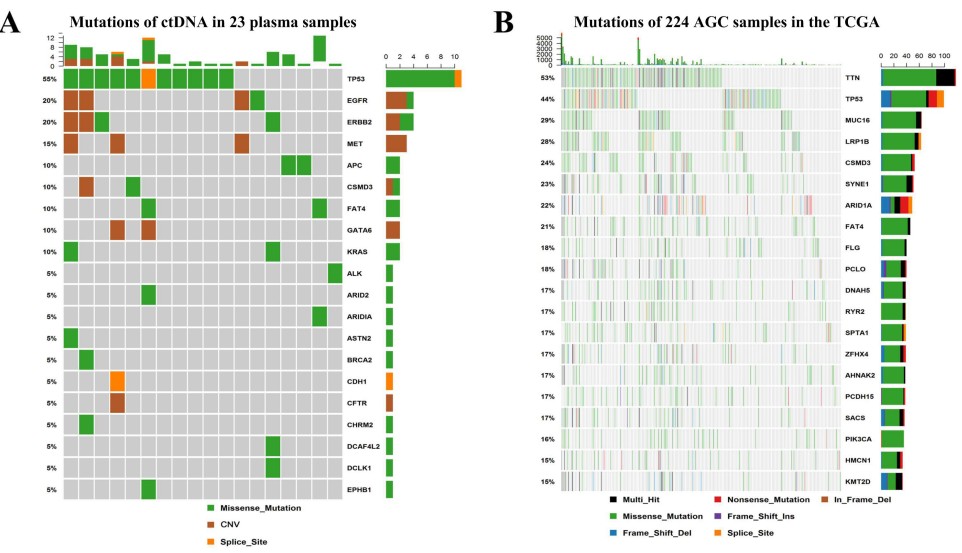

**Figure 2** Landscape of somatic mutations in AGC. (A) The genes and frequencies of ctDNA mutations in 23 plasma samples. (B) The genes and frequencies of 224 AGC sample mutations in the TCGA database.

## Relationship between MET amplification and prognosis of patients with advanced gastric cancer

In order to analyze the effect of MET amplification in tumor tissue on the prognosis of patients with AGC, we downloaded the data with MET amplification information in the TCGA database and found that MET amplification had a relationship with the survival of patients with gastric cancer. Kaplan–Meier survival analysis showed that patients with MET amplification had shorter overall survival than these with MET without amplification (Fig. 4A, $P < 0.001$).

We divided the stomach adenocarcinoma (TCGA) database into early stage gastric cancer (stage I, II) and advanced gastric cancer (stage III, IV). The MET amplification was 1.1% (2/179) in the early stage gastric cancer and 4.6% in the advanced gastric cancer (10/217) ($P = 0.0732$). Therefore, it can be seen that MET amplification was more common in advanced gastric cancer. Kaplan–Meier survival curves showed that disease/progression-free survival (DFS/PFS) of patients with MET amplification in tumor tissue was shorter than the patients with MET without amplification (Fig. 4B, $p = 0.0078$). In the analysis of ctDNA for our 23 plasma samples, Kaplan–Meier analysis showed that patients with MET amplification also had shorter overall survival than these with MET without amplification (Fig. 4C, $P < 0.001$).

## DISCUSSION

Conventional tumor tissue biopsy techniques are not only invasive and harbor risk of complications, but not also suitable for repeated operations. In addition, due to the characteristics of tumor heterogeneity, we cannot obtain accurate tumor molecular profile

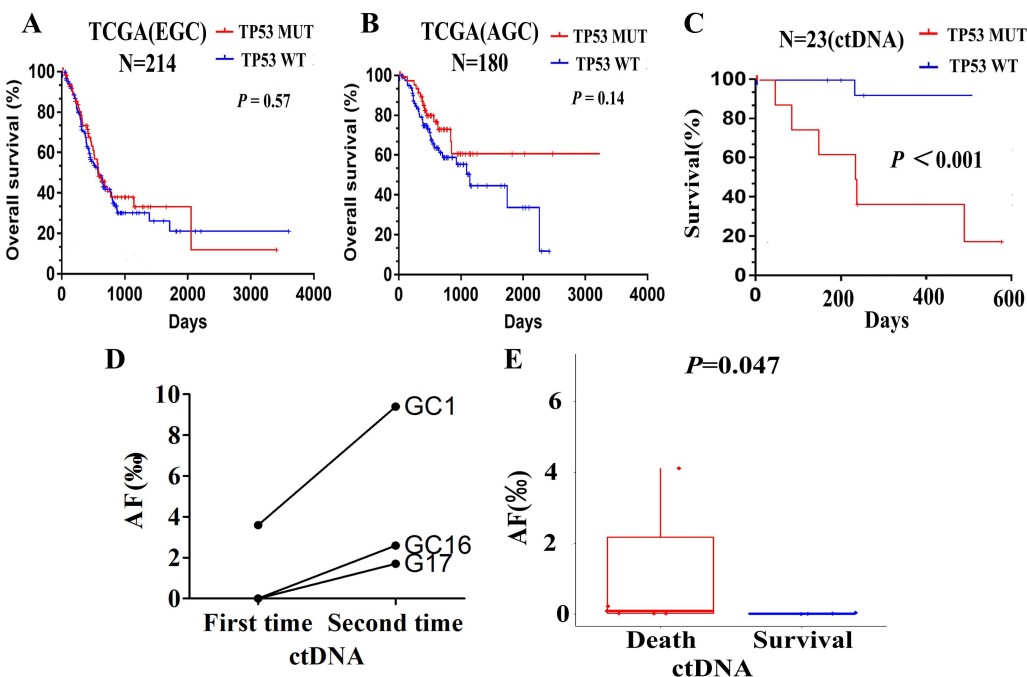

**Figure 3 Relationship between TP53 and prognosis of patients with advanced gastric cancer.** (A) Kaplan-Meier survival analysis of OS based on TP53 status in TCGA database of early stage gastric cancer (stage I, II). (B) Kaplan-Meier survival analysis of OS based on TP53 status in TCGA database of advanced gastric cancer (stage III, IV). (C) Kapla-Meier analysis of survival time based on TP53 status according to the clinical data of AGC patients. (D) The changes of TP53 mutations AF in patient number 1,16 and 17. E. The changes of TP53 mutation AF in patients with survival and death.

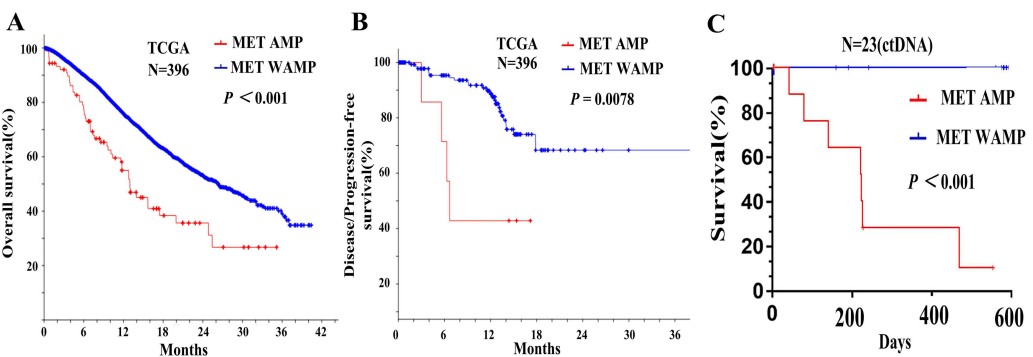

**Figure 4 Relationship between MET amplification and prognosis of patients with advanced gastric cancer.** (A) Kaplan-Meier survival analysis of OS based on MET amplification in TCGA database of gastric cancer. (B) Kaplan-Meier survival analysis of DFS/PFS.

(*Parikh et al., 2019*). Liquid biopsy techniques, such as ctDNA, can not only avoid these limitations, but have also been integrated into daily clinical practice (*Ilie & Hofman, 2016*).

Multigene NGS-panel technology for assessing ctDNA can be used to study a variety of genes, while only a small amount of ctDNA is obtained from even smaller samples,

eventually including circulating tumour cells and free serum DNA (*Meyerson, Gabriel & Getz, 2010*; *Verma et al., 2020*). Comparing with traditional technology (dPCR), NGS sequencing technology has high sensitivity and accuracy for detecting rare ctDNA mutations, thus detecting very rare mutations with AF <0.001% and representing the original DNA population (*Kukita et al., 2015*). Therefore, it is now possible to detect tumor-related mutations on ctDNA and monitor the evolution of cancer progression in patients (*Bettegowda et al., 2014*). Our present report mainly analyzed the evaluation of ctDNA mutations using multigene NGS-panel technology and exploring the clinical significance of TP53 mutation and MET amplification in ctDNA of AGC. We performed mutation detection on ctDNA in 23 patients with AGC and identified the top 20 mutant genes. The five most frequently mutated genes were TP53 (55%), EGFR (20%), ERBB2 (20%), MET (15%) and APC (10%). TP53 was the most common mutated gene in TCGA (44%) and our study (55%). MET amplification had a higher frequency of mutations (15%) in this study, while the frequency of mutations was very low in the TCGA database.

In fact, survival analysis of early gastric cancer or advanced gastric cancer in TCGA database show no significant clinical outcomes in overall survival (OS) based on TP53 status in tumor tissue, which is consistent with previous reports (*Bellini et al., 2012*). We demonstrated a good correlation between the TP53 status of ctDNA and the disease status of AGC. Kaplan–Meier analysis showed that patients with TP53 mutant had shorter overall survival than these with TP53 wild ($P < 0.001$). Since the period of follow-up was set from the blood test time to the end of follow-up, its significance indicated the relationship between the monitoring index (TP53) and the tumor prognosis. We also observed that the AF of TP53 mutation increased after tumor progression, indicating its potential application for monitoring progressive disease. Although TP53 mutation also occurred in surviving patients, it was clearly found that AF was relatively low. Therefore, the results of our study indicated that TP53 mutation might have potential application in predicting disease progression and be related to the AF of TP53 mutations.

Some studies have also investigated the potential of TP53 mutations in ctDNA in gastric cancer. One report demonstrated that ctDNA could be used for monitoring gastric cancer disease progression. Cancer-associated somatic mutations (such as TP53 mutations) are specific to malignancies, and DNA with these mutations indicates the presence of malignancies (*Hamakawa et al., 2015*). The other study examined the role of ctDNA to diagnose or monitor disease status in esophageal and gastric cancer. The allele frequency (AF) of the TP53 mutation was low at diagnosis and rapidly increased during disease progression. (*Kukita et al., 2015*). Focusing on the mutation analysis in GC patients, ctDNA was demonstrated to be a useful predictor for treatment response. The AF of ctDNA decreased after surgical resection and simultaneously increased with recurrent or progressive disease during palliative treatment. A similar trend was seen in some cases, the AF of TP53 mutation remained low until recurrence, at which point the AF increased to 3.6% (*Hamakawa et al., 2015*). Furthermore, in two out of four patients disease recurrence was detected first by ctDNA, with an increasing AF of TP53 earlier before radiological confirmation (*Ueda et al., 2016*). Therefore, these studies demonstrated that TP53 may be

related to the disease progression of GC patients, in a manner that may be related to the AF of TP53 mutations.

In other cancer species, especially in gynecological tumors, the clinical significance and application of TP53 mutations in ctDNA or cfDNA have also been reported. Chiaki Nakauchi et al. demonstrated that the patients with TP53 mutation in cfDNA showed a tendency toward a worse prognosis than those without it (*Nakauchi et al., 2016*). Parkinson et al. retrospectively analyzed TP53 mutations in cfDNA as biomarker of treatment response for patients with relapse high-grade serous ovarian carcinoma and demonstrated that a decrease of ≤60% in TP53 mutant allele fraction after one cycle of chemotherapy was associated with shorter time to progression (*Parkinson et al., 2016*). *Kim et al. (2019)* also found that TP53 mutations in ctDNA was useful as a non-invasive biomarker of treatment response monitoring in patients with high-grade serous ovarian carcinoma. *Savli et al. (2019)* demonstrated that TP53, EGFR and PIK3CA gene variations observed as prominent biomarkers in breast and lung cancer by plasma cell-free DNA genomic testing and were recommended as useful biomarkers for predictive studies to follow up tumor growth. *Treger et al. (2018)* demonstrated for the first time that ddPCR was an effective method for detection of mutant TP53 in ctDNA from children with anaplastic wilm' tumor, even when there was intratumoral somatic heterogeneity. In conclusion, TP53 mutation in ctDNA is the most common in various tumors, the significant clinical of TP53 mutations impact on future risk stratification and surveillance should be further explored in a larger cohort of patients.

We also further explored the prognostic value of MET amplification in the TCGA database and the ctDNA of AGC patients, Kaplan–Meier survival curves showed that OS and DFS/PFS of GC patients with MET amplification was shorter than the patients with MET without amplification ($P < 0.001$ and $P = 0.0078$). The MET amplification of advanced gastric cancer was 4.6% (10/217) higher than that of early gastric cancer 1.1% (2/179), although not statistically significant ($P = 0.0732$). A systematic review and meta-analysis showed that MET overexpression and gene amplification in tumor tissue were indicators of poor prognosis in patients with gastric cancer (*Peng et al., 2014*). Other data show that 8.3% of patients with metastatic gastric cancer exhibit MET amplification, and at least in the Chinese population, MET amplification is more frequent in patients with recurrent/metastatic gastric cancer (*Lane & Levine, 2010*). These findings suggest that MET amplification is not common and may play a key role in advanced gastric cancer.

The detection of MET amplification in ctDNA is still scarce, and only one study has been reported MET alterations detected in ctDNA so far. The study demonstrated that assessment of MET genomic aberrations by liquid biopsy is feasible and found that MET ctDNA anomalies were associated with bone metastases, multiple genomic alterations, and a worse prognosis, including shorter overall survival and a shorter time to recurrence/metastases (*Ikeda et al., 2018*). In our study, the survival rate of patients with advanced gastric cancer with MET amplification was significantly reduced. Therefore, MET amplification in ctDNA may predict disease progression in patients with advanced gastric cancer in the same way as MET amplification in tumor tissues.

In conclusion, TP53 and MET are the two common frequently mutant genes in ctDNA of AGC patients. TP53 may be related to the disease progression of AGC patients, in a manner that may be related to the AF of TP53 mutations. MET amplification in ctDNA may predict disease progression in patients with AGC in the same way as MET amplification in tumor tissues. This study still has some limitations. Firstly, the sample size for ctDNA testing in this study was limited, a multivariate analysis was not conducted. Secondly, we did not analyze the treatment status of the included cases (such as chemotherapy), although they also had a certain impact on the results of ctDNA. Therefore, the clinical utility of TP53 mutation and MET amplification in ctDNA impact on future risk stratification and surveillance should be further explored in a larger cohort of patients.

### Funding

The research was supported by the Ministry of Science and Technology National Key Research and Development Program (2017YFC1700605), the National Health and Family Planning Commission's "Major New Drug Creation" Science and Technology Major Project (2017ZX09304-001), the National Traditional Chinese Medicine Administration National TCM Clinical Research Base Business Construction Research Project (JDZX2015068) and the Key R&D and promotion projects in Henan Province (212102310342). The funders had no role in study design, data collection and analysis, decision to publish, or preparation of the manuscript.

### Grant Disclosures

The following grant information was disclosed by the authors:
Ministry of Science and Technology National Key Research and Development Program: 2017YFC1700605.
National Health and Family Planning Commission's "Major New Drug Creation" Science and Technology Major Project: 2017ZX09304-001.
National Traditional Chinese Medicine Administration National TCM Clinical Research Base Business Construction Research Project: JDZX2015068.
Key R&D and promotion projects in Henan Province: 212102310342.

### Competing Interests

The authors declare there are no competing interests.

### Author Contributions

- Jia Li, Zhaoyan Li, Yajie Ding and Yan Xu conceived and designed the experiments, performed the experiments, analyzed the data, prepared figures and/or tables, authored or reviewed drafts of the paper, and approved the final draft.
- Xiaohong Zhu, Nida Cao, Chen Huang, Mengmeng Qin and Feng Liu conceived and designed the experiments, performed the experiments, analyzed the data, prepared figures and/or tables, and approved the final draft.

- Aiguang Zhao conceived and designed the experiments, analyzed the data, prepared figures and/or tables, authored or reviewed drafts of the paper, and approved the final draft.

## Human Ethics

The following information was supplied relating to ethical approvals (i.e., approving body and any reference numbers):

This study was approved by the Ethics Committee of Shanghai University of Traditional Chinese Medicine (Shanghai, China) and was consistent with the Helsinki Declaration. The reference number for the ethics approval statement is 2017LCSY021.

## DNA Deposition

The following information was supplied regarding the deposition of DNA sequences:

The described sequences are available in the Supplemental Files and at GenBank: SAMN16376921, SAMN16376922, SAMN16376923, SAMN16376924, SAMN16376925, SAMN16376926, SAMN16376927, SAMN16376928, SAMN16376929, SAMN16376930, SAMN16376931, SAMN16376932, SAMN16376933, SAMN16376934, SAMN16376935, SAMN16376936, SAMN16376937, SAMN16376938, SAMN16376939 and SAMN16376940.

## Data Availability

Raw data are available in the Supplemental Files.

## Supplemental Information

Supplemental information for this article can be found online at http://dx.doi.org/10.7717/peerj.11146#supplemental-information.

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
