# Peer review of "TP53 mutation and MET amplification in circulating tumor DNA analysis predict disease progression in patients with advanced gastric cancer"

_PeerJ, doi:10.7717/peerj.11146_

## Round 0.1 · original submission · Major Revisions

Both reviewers were agreed that this study was novel and of potential importance to the field of precision medicine. However, there are concerns over the clarity of the methodology, as well as the clarity and presentation of the data.

Some important issues to be addressed are: What is the relationship between the tumour from the 4 deceased AGC patients and the subsequent ctDNA study? It appears that there is no matched ctDNA for the patient tissue samples, so there are limitations in not being able to directly compare mutations in tissue vs ctDNA in the same patients. Two different NGS panels were also used. There also seem to be discrepancies in the numbers of samples taken from the TGCA database and it is not clear how the results from the 4 patient tissue analyses were cross-compared to the TCGA data (Table 2 is not clear, and there is no clear methodology listed). Please see the reviewers' reports for details of their concerns with the current manuscript and address their comments in any resubmission.

I think it is especially important to address the comments regarding the completeness and clarity of your methodology and the inconsistencies highlighted by the reviewers regarding the datasets and how they are described. It is also important to provide more detail about the statistical analysis.

·

Basic reporting

The study aimed to evaluate the mutational profile of circulating tumor DNA in 23 patients with advanced gastric cancer. For this, the authors used the massive sequencing of this material, in addition to evaluating 4 tumors. Furthermore, they used TCGA data to reveal the agreement between findings.
The authors present a concise and scientific writing, but I suggest a refined revision to find typing and punctuation errors, example “pane”, instead of “panel” and final points in the sentences.
In the introduction, I suggest that the authors expand the topic of epidemiology in the first paragraph, since according to the data shown, China accounts for approximately 68% of all cases of gastric tumors in the world. Are there population factors, such as dietary factors that may influence this percentage?
Table 1 - the authors need to provide the most detailed clinical data, such as stage, chemotherapy cycle. What did the authors consider for the tumor margin study? If necessary, provide as a supplement.
Line 146 – “We performed mutation detection on ctDNA in 23 AGC patients and identified the top 20 mutant genes 147 (Figure 2A)” The authors could provide a figure of the sequencing of the four cases analyzed.

Observation: Are the links to the raw data working?

Experimental design

The study is original and extremely relevant to precision medicine, however, some points in the experimental design have weaknesses that can be improved in their description.

Line 83 - “All patients received written informed consent for this study. The 4 paraffin-embedded (FFPE)...clinical characteristics of 27 AGC patients was also collected (Supplemental Material 1, Sheet 1 and 87 Sheet 2).”
The authors' proposed methodology is not clear. Did the authors intend to compare tissue with the corresponding ctDNA? Why for 5 (n = 28) cases, two collections were made and not for the others? Although this will only be revealed in the results, this information needs to be in methods.
Clinical characteristics of 27 cases are collected. It is not clear why a case was excluded. Wouldn't the data be from the 23 cases included in the study? That way, there would be 28 regardless of more collections.

Line 90 - “DNA from blood by QIAamp DNA Blood Mini Kit and DNA from FFPE tissues by GeneRead DNA FFPE Kit 91 (Qiagen, Hilden, Germany) were extracted according to the manufacturer’s instructions”.
Blood DNA? Or plasma DNA?

Line 99 – “The pane l used in paraffin-embedded (FFPE) tumor tissues was the coding sequence encompassing frequently mutated genes in solid tumors. The pane 2 applied in peripheral blood plasma samples from patients, were designed to cover 471 hot-spot mutations of 197 genes…” The choice of two different panels to compare biological specimens that are paired, seems to me a technical error. If the 450 genes are not covered by panel 2 of 471 genes, they are isolated studies, not comparisons. This given choice of experimental design needs to be better clarified.

Line 121 – “Then, we compared the obtained sequencing data with the genetic sequencing data of patients with advanced gastric cancer in TCGA”. Provide the number of TCGA cases analyzed. Place the number in figure 1.

Validity of the findings

The results of the study are well worked out, but need to be further detailed and explained.

Result
Line 141 - NGS sequencing analysis on paraffin-embedded (FFPE) tumor tissues of 4 dead AGC patients and ctDNA sequencing results of 23 AGC patients were included in Supplemental Material 3. The tumor tissues sequencing data of 4 dead AGC patients and the sequencing data of AGC in the TCGA database were compared taken intersection (Table 2).” Authors need to better describe these results. In the supplementary material 3 - sheet 3, the identification of the specimen type is only as tissue, there needs to be identification of which patient it is. Furthermore, the clinical characteristics of these cases need to be better detailed, what is their stage, what is the histological classification?
Table 2 is not clear the authors' proposal. The authors need to better describe these results, or make available what was done with the TCGA data in a supplement. Table 2 illustrates that the mutations found in three cases out of the four in the study, 10 mutations were coincident with those found in the TCGA, but how often?
Line 176 – “It could therefore be seen that MET amplification was specific in advanced gastric cancer” This statement is not correct. If there were two cases with MET amplification, there is no exclusivity. This is confirmed by the authors' own data: Line 180 - "In the analysis of ctDNA for our 23 plasma samples, Kapla-Meier analysis showed that patients with MET amplification also had shorter overall survival than these with MET without amplification (Figure 4C, P < 0.001)". Of the 23 cases analyzed by the authors, how many had amplification and how many did not, for MET?

Discussion
Line 186 – “Liquid biopsy technology represented by ctDNA, not only can avoid these limitations, but also have been integrated into daily clinical practice [29]” Please rephrase the sentence, the liquid biopsy is not just represented by cfDNA.
Line 193 – “To the best of our knowledge, our present report is the first prospective study on the evaluation of ctDNA…” See the article, Int J Clin Exp Pathol. 2020; 13 (2): 203–211.

Reviewer 2 ·

Basic reporting

The document is mostly clear, although some part of it could be benefited from a rewriting or rephrasing (as quoted in my comments; attached).

More literature in the introduction section is needed to cover basic aspects of the liquid biopsy and cfDNA early research and current research in the cancer reserach field (included in comments attached).

Some extra data should be needed as per my comments in the pdf document (attached).

The results are relevant to the hypotheses.

Experimental design

To my understanding the manuscript falls within the Aims and Scope of the journal.

The methods are not described with sufficient detail and information. This section requires major improvements, as is not well detailed (multiple comments in the pdf attached).
Patients and Samples have to be explained in more detail; data from database have to be explained in more detail; some technical and analytical aspects should be explained in more detail.

The research seems to have high technical standards, however the whole technical section should be better explained (materials and methods requires major improvements).

Research is easy to replicate, and they provide literature which validate their findings.

Validity of the findings

It is an important research, novel enough and with potential impact in its field; however, it has limitations (stated in the manuscript).

More data is required for a better understanding of the research conducted. And a more well detailed methods section is required for a better understanding of the statistical analysis.

Conclusions are related to original research question, but there are limitations (stated in the manuscript)

Additional comments

The introduction section requires a small background about the liquid biopsy and cfDNA/ctDNA application, citing the most important authors.
Materials and Methods section requires major improvements, with extra data and rewriting or rephrasing in some parts.
Please, find in the document attached all my comments.

Annotated reviews are not available for download in order to protect the identity of reviewers who chose to remain anonymous.

---

## Round 0.2 · Minor Revisions

The reviewers have regarded the changes you have made to the manuscript favourably. There are still some consistency issues regarding the presentation of your methodology and results that should be addressed so that the study is clearer. In their report, reviewer 2 has highlighted several areas for improvement (see their annotated pdf). The crucial aspects that I feel need to be improved are:

Completeness and clarity of methodology
1. Although the information relating to the patients and samples is clearer in the amended text, as reviewer 2 has highlighted it could still be improved. Some information such as the mention of rapid disease progression needs included somewhere in the figures/tables, while the time frame between the repeated blood draws needs to be stated.
2. As the tissue samples are now not being used in the paper, please ensure the methods section removes any mention of them. For example, it still states that NGS libraries were prepared from tissue and germ line.
3. Processing of data – clarify the validation process. It is not currently clear whether your 23 AGC samples were validated by using the TCGA database or if you looked at the database first and validated those findings with the 23 samples. Figure 1 suggests they were looked at in parallel.

Inconsistencies in reporting of results
1. ‘Landscape of mutations’ results section – you state this is for 23 samples, so were the repeat samples not included in this landscape analysis?
2. You state that 5 patients had rapid disease progression and were sequenced twice, but only patients 1 and 16 are reported. What about the other 3? Did they not show changes? It is important to include all patients and not exclude any just because they didn’t show changes. Additionally, the allele frequencies stated in the text (0.36%, 0.94%, 0.26%) do not match that plotted on the graph in Figure 3D (26%, 94%, 36%). There is a huge difference between an allele frequency of 0.94% and 94%, so please correct either the text or figure as appropriate. What was the threshold used for counting a patient as TP53 mutant in the survival analysis (any presence of mutant?)
3. In your conclusion “We also observed that the AF of TP53 mutation increased after tumor progression, indicating its potential application for monitoring progressive disease.” – was this true for all 5 patients that were re-tested following tumour progression?
4. Ensure supplemental material contains what it is stated to contain e.g. supplemental material 2 doesn’t have coordinates of selected genes, even though this is stated in the materials and methods.

Consistency in citations (fit to journal format)
Please ensure your references follow the required journal format.
When referring to work by specific groups, usually only the surname is needed, not both names e.g. Hanakawa et al reported….. rather than T. Hanakawa. There are other examples that also need fixed. Ensure references are accurate and your text accurately reflects what each paper reports e.g. Reviewer 2 has queried your reference to a Pan et al paper in the conclusion.

·

Basic reporting

The authors made all corrections regarding grammatical and typographical errors.
They added new references throughout the text, especially in the introduction, making the manuscript more complete.
Suggestions for changes to the figures have been accepted and all are easy to view.

Experimental design

The authors made all corrections regarding my comments and suggestions. They excluded sentences and data that raised doubts about the study's methodology. After the corrections, the data became more technical and easier to interpret.

Validity of the findings

After all corrections and changes in the text, the manuscript is clear and more scientifically robust.
The methodology and statistics used are adequate.

Additional comments

Congratulations on your study and thanks for accepting the suggested changes.

Reviewer 2 ·

Basic reporting

The manuscript has been widely modified for the previous version, and in general terms is better now.
Introduction is well presented, highlighting the importance and interest of the study being carried, including multitude of literature reference that clearly back their research.
Multiple tables and figures have been introduced, making it easier to intrepret the manuscript and the data presented in it.

Experimental design

The experimental design is clearly stated and explained; however, there are major discrepancies regarding the amount of samples and the test run on some of the samples.
The major issue regarding samples is that it is mentioned in the manuscript that there are 23 patients, but 28 samples in total. For 5 of those 23 patients an extra blood sample was taken.
However, there are no information about those 5 patients, the extra blood sample, the time between the first and the second blood sample, the NGS analysis being carried over the 23 or the 28 total samples, ... Multiples discrepancies not resolved in the results section, and vaguely mentioned in the methods section.

Validity of the findings

Findings are interesting and surely match on the current need of finding new cancer biomarkers and progress into a more personalized medicine.
The amounnt of recent literature provided by the authors clearly showed that multiple efforts by different teams are being directed now to the identification of new and more effective cancer biomarkers in gastric cancer. This research fit within those efforts.
Conclusions and discussion were very clear.

Annotated reviews are not available for download in order to protect the identity of reviewers who chose to remain anonymous.

---

## Round 0.3 · accepted · Accept

Your revised manuscript addresses the majority of the suggested revisions and is suitable for publication. There are 2 minor things that should be clarified in the final version.

1. Pg 12 - The Pan et al reference previously flagged up by reviewer 2, refers to sequencing of FFPE tissue, but the conclusion uses it as a reference for ctDNA. Please amend the sentence to more accurately reflect this reference or use an alternative reference.

2. Pg 14 - The 2 sentences prior to the Kukita et al reference are contradictory. The first says TP53 allele frequency increased, the second says there was no significant change. Please clarify this in the text.